Kindia (Pavetteae, Rubiaceae), a new cliff-dwelling genus with chemically profiled colleter exudate from Mt Gangan, Republic of Guinea

Cheek Martin 1
Magassouba Sékou 2
Howes Melanie-Jayne R. 3
Doré Tokpa 2
Doumbouya Saïdou 4
Molmou Denise 2
Grall Aurélie 1
Couch Charlotte 1
Larridon Isabel i.larridon@kew.org 1 5
1 Identification and Naming, Royal Botanic Gardens Kew , Richmond , Surrey , United Kingdom
2 Herbier National de Guinée, Université de Gamal Abdel Nasser de Conakry , Conakry , République de Guinée
3 Natural Capital and Plant Health, Royal Botanic Gardens Kew , Richmond , Surrey , United Kingdom
4 Centre d’Observation de Surveillance et d’Informations Environnementales, Ministère de l’Environnement et des Eaux et Forêts , Conakry , Guinea-Conakry
5 Department of Biology, Research Group Spermatophytes, Ghent University , Ghent , Belgium
Scheibe Renate
Electronic publication date: 2018 Apr 20
Publication date: 2018
Volume: 6
Electronic Location ID: e4666
Received 2017 Dec 2; Accepted 2018 Apr 4
Copyright: ©2018 Cheek et al.
Copyright year: 2018
Copyright holder: Cheek et al.
License: This is an open access article distributed under the terms of the Creative Commons Attribution License, which permits unrestricted use, distribution, reproduction and adaptation in any medium and for any purpose provided that it is properly attributed. For attribution, the original author(s), title, publication source (PeerJ) and either DOI or URL of the article must be cited.
License URL: https://creativecommons.org/licenses/by/4.0/

Keywords: Cliff-dwelling, Conservation, Guinea-conakry, Epilithic, Tropical Important Plant Areas, Rubiaceae

Funding: B.A. Krukoff fund Darwin Initiative of the Department of the Environment Food and Rural Affairs (DEFRA) Professor Isabel Larridon is supported by the B.A. Krukoff fund. Rio Tinto funded the mission to the Republic of Guinea during which this discovery was made as part of their contribution to biodiversity capacity building and redlisting which contributes to the ‘Important Plant Areas in the Republic of Guinea’ supported by the Darwin Initiative of the Department of the Environment Food and Rural Affairs (DEFRA), UK government (project Ref. 23–002). The funders had no role in study design, data collection and analysis, decision to publish, or preparation of the manuscript.

==============================
A new genus Kindia (Pavetteae, Rubiaceae) is described with a single species, Kindia gangan, based on collections made in 2016 during botanical exploration of Mt Gangan, Kindia, Republic of Guinea in West Africa. The Mt Gangan area is known for its many endemic species including the only native non-neotropical Bromeliaceae Pitcairnia feliciana. Kindia is the fourth endemic vascular plant genus to be described from Guinea. Based on chloroplast sequence data, the genus is part of Clade II of tribe Pavetteae. In this clade, it is sister to Leptactina sensu lato (including Coleactina and Dictyandra). K. gangan is distinguished from Leptactina s.l. by the combination of the following characters: its epilithic habit; several-flowered axillary inflorescences; distinct calyx tube as long as the lobes; a infundibular-campanulate corolla tube with narrow proximal section widening abruptly to the broad distal section; presence of a dense hair band near base of the corolla tube; anthers and style deeply included, reaching about mid-height of the corolla tube; anthers lacking connective appendages and with sub-basal insertion; pollen type 1; pollen presenter (style head) winged and glabrous (smooth and usually hairy in Leptactina); orange colleters producing a vivid red exudate, which encircle the hypanthium, and occur inside the calyx and stipules. Kindia is a subshrub that appears restricted to bare, vertical rock faces of sandstone. Fruit dispersal and pollination by bats is postulated. Here, it is assessed as Endangered EN D1 using the 2012 IUCN standard. High resolution LC-MS/MS analysis revealed over 40 triterpenoid compounds in the colleter exudate, including those assigned to the cycloartane class. Triterpenoids are of interest for their diverse chemical structures, varied biological activities, and potential therapeutic value.

Introduction

Plant conservation priorities are often poorly represented in national and global frameworks due to a lack of publicly available biodiversity data to inform conservation decision-making (Corlett, 2016; Darbyshire et al., 2017), despite the fact that one in five plant species are estimated to be threatened with extinction mainly due to human activities (Brummitt et al., 2015; Bachman et al., 2016). West Africa represents a priority target area for future efforts in botanical exploration to inform conservation action and biological resource use (Sosef et al., 2017).

Botanical exploration and new species discovery in Guinea

Guinea has numerous endemic species and a high diversity of species in the context of West Tropical African countries (c. 3,000 species; Lisowski, 2009), including several endemic genera, i.e., Fleurydora A.Chev. (Ochnaceae), Feliciadamia Bullock (Melastomataceae), Cailliella Jacq.-Fél. (Melastomataceae). Botanical exploration, discovery and publication of new species appeared to have nearly stopped after Independence in 1958, with the exception of the work carried out by S Lisowski (1924–2002). His work resulted in the publication of several new species, e.g., Pseudoprosopis bampsiana Lisowski, Mikaniopsis camarae Lisowski and Bacopa lisowskiana Mielcarek, and the posthumously published ‘Flore de la République de Guinée’ (Lisowski, 2009). The other species new to science that were published in the period 1960–2010 were based on specimens collected in the French Colonial period, e.g., Phyllanthus felicis Brunel (1987) and Clerodendrum sylvae Adam (1974). In recent years, this has begun to change as botanical exploration, often associated with environmental impact assessments for more environmentally responsible mining companies such as Rio Tinto (Harvey et al., 2010; Magassouba et al., 2014), has restarted. Xysmalobium samoritourei Goyder (2009), Gymnosiphon samoritoureanus Cheek & Van der Burgt (2010), Eriosema triformum Van der Burgt et al. (2012), Brachystephanus oreacanthus Champluvier & Darbyshire (2009), Striga magnibracteata Fischer, Darbyshire & Cheek (2011), Isoglossa dispersa Darbyshire, Pearce & Banks (2012), Eriocaulon cryptocephalum Phillips & Mesterházy (2015), Napoleonea alata Prance & Jongkind (2015) and Psychotria samouritourei Cheek & Williams (2016) are examples of recent new discoveries from Guinea resulting from this impetus. Just across the border in Mali, Calophyllum africanum Cheek & Luke (2016) was recently found, and in Ivory Coast Macropodiella cussetiana Cheek & Ameka (2016). Even a new rheophytic genus, Karima Cheek & Riina has come to light in Guinea (Cheek et al., 2016). Many of the new species being described are narrow endemics and are threatened by habitat clearance for subsistence agriculture, open-cast mining, urban expansion, quarrying (Couch et al., 2014) and invasive species (Cheek et al., 2013).

Mt Gangan: a Tropical Important Plant Areas

The criteria of the Important Plant Areas (IPAs) programme, developed by Plantlife International (2004), offers a pragmatic yet scientifically rigorous means of delivering biodiversity datasets, enabling informed site-based conservation priorities (Darbyshire et al., 2017). IPAs are aligned to Target 5 of the Convention on Biological Diversity (CBD)’s ‘Global Strategy for Plant Conservation’ and so offer an important step toward fulfilling national CBD targets (Darbyshire et al., 2017). IPAs are identified on the basis of three criteria: the presence of threatened species, exceptional botanical richness and threatened habitats (Anderson, 2002; Plantlife International, 2004). These criteria were recently revised for a global approach (Darbyshire et al., 2017), and are used in the Tropical Important Plant Areas programme of the Royal Botanic Gardens, Kew. In Guinea, botanical exploration is used to aid in aligning the existing forest reserve network, which focuses on maintaining timber resources for exploitation, and the existing few National Parks protecting large mammals or wetlands, to cover global priority areas for plant conservation.

The Mt Gangan area was identified as a prospective Tropical Important Plant Area (Larridon & Couch, 2016; Herbier National de Guinée, 2017; Darbyshire, continuously updated). Mt Gangan is an outlier of the Fouta Djallon Highlands of Guinea, and is an area of sandstone table mountains with sheer cliffs, frequent rock ledges, overhangs and caves. The rock formations create a variety of microhabitats and are inhabited by sparse small trees, shrubs, subshrubs and perennial herbs, many of which are rock specialists, such as Fegimanra afzelii Engl. Fleurydora felicis A.Chev., Clerodendrum sylvae, Phyllanthus felicis, Cyanotis ganganensis R.Schnell, Dissotis pygmaea A.Chev. & Jacq.-Fél., Dissotis humilis A.Chev. & Jacq.-Fél. and Melastomastrum theifolium (G.Don) A.Fern. & R.Fern var. controversum (A.Chev. & Jacq.-Fél.) Jacq.-Fél. (formerly Dissotis controversa (A.Chev. & Jacq.-Fél.) Jacq.-Fél.). Except for Fegimanra afzelii, the abovementioned species are all either endemic or near-endemic to the Mt Gangan complex of precipitous sandstone table mountains. Mt Gangan is also home to Pitcairnia feliciana (A. Chev) Harms & Mildbr., the only non-neotropical Bromeliaceae (Porembski & Barthlott, 1999).

A new Rubiaceae from Mt Gangan

In February 2016, a survey was initiated of the vegetation types, plant species, and threats at Mt Gangan. During the survey an unusual Rubiaceae was observed with more or less sessile leaf rosettes (Cheek 18345), growing only on vertical faces of bare sandstone cliffs that form the flanks of parts of some of the sandstone table mountains that comprise Mt Gangan (Fig. 1). Cheek 18345 has fruits (Fig. 1) and only old, dried flowers. Because the old flowers were mistakenly interpreted as likely to have had valvate corolla aestivation, and because the inflorescences were axillary, with two-celled, fleshy fruits, containing numerous seeds, the species was initially placed in tribe Mussaendeae sensu Hepper & Keay (1963: 104), using the key to the tribes of Rubiaceae in the Flora of West Tropical Africa. Within this tribe, it keyed out as Sabicea Aubl. However, it matched no known species of that genus, being bizarre in several features, such as the epilithic habit, the red colleter exudate, and the seeds with a central excavation. Checks with all other genera of Rubiaceae in West Tropical Africa, and indeed tropical Africa, also produced no matches, leading to the hypothesis that this taxon represented a new genus to science. In June and September 2016, additional specimens (Cheek 18541A and Cheek 18602) of the taxon were obtained during the flowering season, at which time the corolla aestivation was found to be contorted to the left (Fig. 1), excluding it from Sabicea but consistent with Pavetteae (De Block et al., 2015), as was first indicated by the results of the molecular study (see below). However, the axillary inflorescences are unusual in that tribe (De Block et al., 2015). In this study, morphological and chloroplast sequence data are employed to test the hypothesis that the new Rubiaceae from Mt Gangan is: (1) part of tribe Pavetteae, and (2) represents a new genus to science. To achieve this, we aim to investigate the overall morphology and the pollen morphology and compare them to those found in other tribe Pavetteae genera, and place the taxon in a molecular phylogenetic framework of the tribe. Ecology and conservation status of the new Rubiaceae are also investigated, as is the colleter exudate biochemistry because of its unusual red colour.

Figure 1 Photographs showing the cliff-dwelling habitat and the habit of K. gangan at Mt Gangan, Kindia, Guinea.

(A) plants scattered on high sandstone cliff (Cheek 18345); (B) plant habit on cliff face (Cheek 18541A); (C) frontal view of flower (Cheek 18541A); (D) side view of inflorescence showing cupular bract (Cheek 18541A); (E) opened fruit showing ripe seeds (Cheek 18345). Photos taken by Martin Cheek.

Materials and Methods

Ethics statement

The specimens studied were collected as a part of field surveys for the ‘Important Plant Areas in the Republic of Guinea’ project funded by a Darwin Initiative grant of the Department of the Environment, Food and Rural Affairs (DEFRA) of the government of the United Kingdom. Permits to export these specimens were issued by the Ministère de l’Environnement et des Eaux et Forêts of the Republic of Guinea, Certificat d’Origine no 0000344 (date 21 June 2016) and no 0000399 (dated 28 October 2016). Specimens were collected under the terms of a Memorandum of Understanding between the Board of Trustees, RBG, Kew and the Herbier National de Guinée, Université Gamal Abdel Nasser de Conakry, renewed and extended for 5 years in December 2015. The study area at Mt Gangan reported in this paper is controlled by the government of the Republic of Guinea and is not privately owned, nor protected. The taxon studied here is not yet a protected species.

Taxonomy

The electronic version of this article in Portable Document Format (PDF) will represent a published work according to the International Code of Nomenclature for algae, fungi, and plants (ICN), and hence the new names contained in the electronic version are effectively published under that Code from the electronic edition alone. In addition, new names contained in this work which have been issued with identifiers by IPNI will eventually be made available to the Global Names Index. The IPNI LSIDs can be resolved and the associated information viewed through any standard web browser by appending the LSID contained in this publication to the prefix “http://ipni.org/”. The online version of this work is archived and available from the following digital repositories: PeerJ, PubMed Central, and CLOCKSS.

Morphological study

Herbarium material was examined with a Leica Wild M8 dissecting binocular microscope fitted with an eyepiece graticule measuring in units of 0.025 mm at maximum magnification. The drawing was made with the same equipment with a Leica 308700 camera lucida attachment. For dissection, structures were first rehydrated by soaking in water with surfactant. The overall morphology was documented, described and illustrated following botanical standard procedures (Davis & Heywood, 1963). Information about habit, habitat, and distribution was taken from specimen labels and field observations.

Material of Cheek 18345, Cheek 18529, Cheek 18541A and Cheek 18602, the new Rubiaceae of Mt Gangan, was first compared morphologically against reference material of all Pavetteae genera held at K. The study was then extended to include the BM, HNG, P and WAG herbaria. Codes for cited herbaria follow Index Herbariorum (Thiers, continuously updated). The main online search address used for retrieving specimen data from P (which globally has the largest holdings of herbarium specimens from the Republic of Guinea) was https://science.mnhn.fr/institution/mnhn/collection/p/item/p00179355?listIndex=128&listCount=610; that for WAG was http://bioportal.naturalis.nl/geographic-search?language=en. Special focus was given to taxa shown to be closely related by the molecular phylogenetic results. All specimens marked ‘!’ have been seen.

Pollen morphology has been shown to be useful in characterising clades, and sometimes genera within tribe Pavetteae (De Block & Robbrecht, 1998). Pollen samples were collected from Cheek 18541A (K). Whole, unacetolysed anthers were placed on a stub using double-sided tape and sputter-coated with platinum in a Quorom Q150T coater for 30 s and examined in a Hitachi 54700 scanning electron microscope at an acceleration voltage of 4 kV.

Molecular methods

In this study, previously published chloroplast sequence data was used (De Block et al., 2015), supplemented with new sequences from selected regions (rps16 and trnT-F) (Appendix). The DNA extraction protocol and material and methods for amplification and sequencing used in this study follow De Block et al. (2015).

Sequences were assembled and edited in Geneious R8 (http://www.geneious.com; Kearse et al., 2012), aligned using MAFFT 7 (Katoh, Asimenos & Toh, 2009; Katoh & Standley, 2013); afterwards, alignments were checked manually in PhyDE 0.9971 (Müller et al., 2010). The alignments used to produce the phylogenies are available as Data S1.

Based on De Block et al. (2015), the alignments of the two chloroplast regions were concatenated for the downstream analyses, each marker was treated as a separate partition, and both partitions were analysed using the GTR + G model. Maximum likelihood (ML) analyses were performed using RAxML 8.2.10 (Stamatakis, 2014). The search for an optimal ML tree was combined with a rapid bootstrap analysis of 1,000 replicates. Bayesian Inference (BI) analyses were conducted in MrBayes 3.2.6 (Ronquist et al., 2012). Rate heterogeneity, base frequencies, and substitution rates across partitions were unlinked. The analysis was allowed to run for 100 million generations across 4 independent runs with four chains each, sampling every 10,000 generations. Convergence, associated likelihood values, effective sample size values and burn-in values of the different runs were verified with Tracer 1.5 (Rambaut et al., 2014). The first 25% of the trees from all runs were excluded as burn-in before making a majority-rule consensus of the 30,000 posterior distribution trees using the “sumt” function. All phylogenetic analyses were run using the CIPRES portal (http://www.phylo.org/; Miller, Pfeiffer & Schwartz, 2010). Trees were drawn using TreeGraph2 (Stöver & Müller, 2010) and FigTree 1.4.3 (Rambaut, 2016), and adapted in Adobe Photoshop CS5.

Ecology and conservation status

Field studies were conducted in the Mt Gangan complex north of Kindia in February (fruiting season), June and September (flowering season) 2016, and in November 2017 (fruiting season). Plants of the new taxon were mostly inaccessible on vertical sandstone cliffs, so they were studied and counted with binoculars. Voucher specimens were made in the usual way (Bridson & Forman, 1998) from the few accessible plants that could be reached from the base of the cliffs. The conservation assessment was prepared following IUCN (2012) with the help of Bachman et al. (2011). The distribution of the species was mapped using SimpleMappr (Shorthouse, 2010).

LC-MS/MS analysis of colleter exudate

A sample of Cheek 18345 was prepared by extracting the colleter exudate fragments in EtOH:MeOH:H2O (5:4:1) (1 mg/ml) for 24 h, prior to centrifugation. The supernatant was then subjected to LC–MS/MS analysis. Analyses were performed on a Thermo Scientific system consisting of an ‘Accela’ U-HPLC unit with a photodiode array detector and an ‘LTQ Orbitrap XL’ mass spectrometer fitted with an electrospray source (Thermo Scientific, Waltham, MA, USA). Chromatography was performed with a 5 µl sample injection onto a 150 mm × 3 mm, 3 µm Luna C-18 column (Phenomenex, Torrance, CA, USA) using the following 400 µl/min mobile phase gradient of H2O/CH3CN/CH3CN +1% HCOOH: 90:0:10 (0 min), 0:90:10 (20 min), 0:90:10 (25 min), 90:0:10 (27 min), 90:0:10 (30 min). The ESI source was set to record high resolution (30 k resolution) MS1 spectra (m/z 125–2,000) in negative mode and data dependent MS2 and MS3 spectra using the linear ion trap. Detected compounds were assigned by comparison of accurate mass data (based on ppm), and by available MS/MS data, with reference to the published compound assignment system (Schymanski et al., 2014).

Results

Morphology

Characters separating the new Rubiaceae from Mt Gangan from its sister genus Leptactina are provided in Table 1. A detailed description is given in the taxonomic treatment below.

The pollen grains (Fig. 2) are tricolporate, overall spheroidal, but usually triangular in polar view 20–25 µm in diameter, with an apocolpium of 3.5–4.5 µm diameter, giving an apocolpial index of 0.125. The mesocolpium sculpturing is microperforate- reticulate, the reticulum units are obscurely pentagonal, about 900–1,000 nm in diameter, the muri broad and rounded, the central perforations c. 0.1 µm. The apocolpium exine sculpturing grades to microporate. The colpi are about 4–6 µm wide at the equator, 2 µm wide at the poles. The colpal membrane is densely granular, the granular units 0.2–0.5 µm diameter, the margin with the mesocolpium well-defined but irregular, and the pores 3–5 µm in diameter.

Figure 2 Scanning electron micrographs of triangular pollen (unacetolysed) of K. gangan.

(A) polar view; (B) surface sculpturing. From Cheek 18541A.

Table 1 Characters separating Kindia from Leptactina s.l., including Coleactina and Dictyandra (i.e., the remainder of Pavetteae Clade II according to De Block et al., 2015).

Data for Leptactina morphology were taken from specimen measurements and from Hallé (1970) and Neuba, Malan & Kouadio (2014). Data for the pollen characters of Leptactina are based on De Block & Robbrecht (1998).

Characters	Leptactina s.l.	Kindia	
Pollen: apocolpial index	0.39–0.68	0.125	
Pollen aperture number	(3–)4	3	
Anther attachment	Sub-apical or medifixed (except sub-basal in L. arborescens)	Sub-basal	
Anther apical connective appendage	Present	Absent	
Style arms at anthesis	Divergent (except L. pynaertii)	Appressed together	
Corolla tube shape	Long narrow cylindrical sometimes widening subtly at the throat (where anthers are included)	Strongly infundibular-campanulate, short proximal narrow section abruptly widening to long, broad distal section	
Presence of a dense, discrete band of hairs near base of corolla tube	Absent	Present	
Pollen presenter	Smooth, usually hairy	Longitudinally winged, glabrous	
Colleter exudate from apical bud	Usually not conspicuous; if conspicuous, translucent, colourless	Conspicuous, opaque, bright red	

Molecular phylogeny

The concatenated ML and BI analyses did not significantly differ in topology, therefore the results discuss the relationships shown in the majority consensus multiple-locus BI tree with the associated posterior probability (PP) values and the bootstrap (BS) values of the multiple-locus ML tree (Fig. S1), and summarised in Fig. 3. As the data used here is largely based on the dataset used by De Block et al. (2015), the relationships recovered here largely match those published in that study. Within a well-supported tribe Pavetteae (BS = 100, PP = 1), four major clades (I–IV) were retrieved. However, although in De Block et al. (2015) Clade I was retrieved as sister to a polytomy of Clades II–IV, in this study Clade I + III (BS = 90, PP = 0.99) and Clade II + IV (BS = 79, PP = 0.87) are supported as separate clades. Clade I (BS = 100, PP = 1) included the African genera Nichallea Bridson and Rutidea DC. Clade II (BS = 100, PP = 1) comprised the African genus Leptactina Hook.f. sensu De Block et al. (2015) and the new Rubiaceae from Mt Gangan, with the latter sister to Leptactina of which the monophyly is well supported (BS = 99, PP = 1). Clade III (BS = 87, PP = 0.87) consisted of the paleotropical genus Pavetta L., the monotypic East African genus Cladoceras Bremek. and the African species of Tarenna Gaertn. In our BI analysis, the species Tarenna jolinonii N.Hallé was recovered as sister to the rest of a weakly supported Clade III, as was found in the results of De Block et al. (2015). However, in the ML analysis, this species was weakly supported as sister to Clade I. Clade IV (BS = 92, PP = 1) included the East African monotypic genus Tennantia Verdc., Asian/Pacific and Madagascan species of Tarenna, the Madagascan endemics Homollea Arènes, Robbrechtia De Block and Schizenterospermum Homolle ex Arènes and the Afro-Madagascan genera Paracephaelis Baill. and Coptosperma Hook.f. As in the results of De Block et al. (2015), the nodes in this clade were poorly supported and the relationships between subclades remained unclear.

Figure 3 Summary phylogenetic hypothesis based on the concatenated BI analysis.

Clades I–IV were numbered according to De Block et al. (2015).

LC-MS/MS analysis of colleter exudate

High resolution LC-MS/MS analysis revealed the detection of a range of triterpenoids in the exudate, including those assigned as the cycloartane class (Table 2). This included a compound eluting at the retention time (Rt) 14.3 min with m/z 499.3068 that was assigned the molecular formula C30H44O6 from the observed [M-H]− ion, which is that of dikamaliartane A, or isomer. Four compounds eluting at Rt 23.8, 25.3, 25.9 and 26.9 min were assigned the molecular formula C30H46O4, from their observed [M-H]− ions, which is that of dikamaliartane D, F, or isomer. The cycloartane triterpenoids, dikamaliartanes A, D and F, have previously been reported to occur in dikamali gum, which is the colleter exudate of Gardenia gummifera L.f. and G. resinifera Roth. (Kunert et al., 2009), in the Rubiaceae.

Table 2 Compounds assigned from LC-MS/MS analysis (negative mode) of the colleter exudate from Cheek 18345.

Assigned compound#(or isomer)	Retention time (min)	Molecular formula	(m/z)	Ion	ppm#	
Trihydroxy-oxocycloartanoic acid	12.3	C30H48O6	503.3385	[M-H]−	1.366	
Pentahydroxy-(hydroxylmethyl) cycloartanoic acid	12.4	C31H52O8	551.3596	[M-H]−	1.230	
Epoxy-trihydroxy-cycloartenoic acid	12.9	C30H46O6	501.3228	[M-H]−	1.112	
Epoxy-trihydroxy-cycloartenoic acid	13.0	C30H46O6	501.3225	[M-H]−	0.993	
Epoxy-trihydroxy-cycloartenoic acid	13.2	C30H46O6	501.3231	[M-H]−	1.910	
Epoxy-trihydroxy-cycloartenoic acid	13.3	C30H46O6	501.3229	[M-H]−	1.372	
Trihydroxy-oxocycloartanoic acid	13.8	C30H48O6	503.3379	[M-H]−	0.154	
Trihydroxy-oxocycloartanoic acid	14.0	C30H48O6	503.3380	[M-H]−	0.273	
Dikamaliartane Aa	14.3	C30H44O6	499.3068	[M-H]−	0.556	
Trihydroxy-oxocycloartanoic acid	14.6	C30H48O6	503.3384	[M-H]−	1.247	
1,3-Dihydroxy-23-oxocycloartan-28-oic acid (=carinatin A)b or 4,28-dihydroxy-26-oxo-3,4-secocycloart-24-en-3-oic acid (=gardenoin J)c	15.0	C30H48O5	487.3435	[M-H]−	1.195	
1,3-Dihydroxy-23-oxocycloartan-28-oic acid (=carinatin A)b or 4,28-dihydroxy-26-oxo-3,4-secocycloart-24-en-3-oic acid (=gardenoin J)c	15.9	C30H48O5	487.3433	[M-H]−	0.743	
1,3-Dihydroxy-23-oxocycloartan-28-oic acid (=carinatin A)b or 4,28-dihydroxy-26-oxo-3,4-secocycloart-24-en-3-oic acid (=gardenoin J)c	16.3	C30H48O5	487.3432	[M-H]−	0.559	
1,3-Dihydroxy-23-oxocycloart-24-en-28-oic acid (=gardenolic acid B)d	16.5	C30H46O5	485.3274	[M-H]−	0.355	
1,3-Dihydroxy-23-oxocycloartan-28-oic acid (=carinatin A)b or 4,28-dihydroxy-26-oxo-3,4-secocycloart-24-en-3-oic acid (=gardenoin J)c	16.6	C30H48O5	487.3432	[M-H]−	0.682	
1,3-Dihydroxy-23-oxocycloart-24-en-28-oic acid (=gardenolic acid B)d	17.3	C30H46O5	485.3276	[M-H]−	0.746	
1,3-Dihydroxy-23-oxocycloart-24-en-28-oic acid (=gardenolic acid B)d	17.5	C30H46O5	485.3272	[M-H]−	0.016	
1,3-Dihydroxy-23-oxocycloart-24-en-28-oic acid (=gardenolic acid B)d	17.8	C30H46O5	485.3280	[M-H]−	1.550	
Epoxy-trihydroxy-cycloartenoic acid	18.2	C30H46O6	501.3228	[M-H]−	1.292	
1,3-Dihydroxy-23-oxocycloart-24-en-28-oic acid (=gardenolic acid B)d	19.4	C30H46O5	485.3279	[M-H]−	1.303	
1,3-Dihydroxy-23-oxocycloartan-28-oic acid (=carinatin A)b or 4,28-dihydroxy-26-oxo-3,4-secocycloart-24-en-3-oic acid (=gardenoin J)c	19.5	C30H48O5	487.3432	[M-H]−	0.682	
1,3-Dihydroxy-23-oxocycloart-24-en-28-oic acid (=gardenolic acid B)d	19.9	C30H46O5	485.3272	[M-H]−	0.016	
1,3-Dihydroxy-23-oxocycloartan-28-oic acid (=carinatin A)b or 4,28-gihydroxy-26-oxo-3,4-secocycloart-24-en-3-oic acid (=gardenoin J)c	20.3	C30H48O5	487.3434	[M-H]−	0.928	
Gummiferartane 3e	20.8	C30H50O5	489.3549	[M-H]−	0.638	
1,2,3,4-Octadecanetetrol; 1-O-rhamnosidef	20.9	C24H48O8	463.3281	[M-H]−	0.903	
7-Hydroxy-3,4-secocycloarta-4(28),24-diene-3,26-dioic acid; 3-Me ester or 4-hydroxy-3,4-secocycloart-24-en-26,22-olid-3-oic acid; Me ester	21.0	C31H48O5	499.3435	[M-H]−	1.166	
23,26-Epoxy-6,28-dihydroxy-3,4-secocycloarta-4(29),23,25-trien-3-oic acidg	21.2	C30H44O5	483.3124	[M-H]−	1.619	
1,2,3,4-Eicosanetetrolh	21.6	C20H42O4	391.3069	[M + HCOO]−	0.863	
Gummiferartane 3e	21.8	C30H50O5	489.3590	[M-H]−	0.883	
1,3-Dihydroxy-23-oxocycloartan-28-oic acid (=carinatin A)b or 4,28-dihydroxy-26-oxo-3,4-secocycloart-24-en-3-oic acid (=gardenoin J)c	22.0	C30H48O5	487.3433	[M-H]−	0.805	
1,2,3,4-Octadecanetetrol; 1-O-rhamnosidef	22.4	C24H48O8	463.3283	[M-H]−	1.378	
1,2,3,4-Octadecanetetrol; 1-O-rhamnosidef	22.5	C24H48O8	463.3283	[M-H]−	1.443	
1,3-Dihydroxy-23-oxocycloartan-28-oic acid (=carinatin A)b or 4,28-dihydroxy-26-oxo-3,4-secocycloart-24-en-3-oic acid (=gardenoin J)c	22.8	C30H48O5	487.3435	[M-H]−	1.318	
Dihydroxy-methoxycycloartenoic acid or diepoxy-methoxycycloartane-diol	23.0	C31H50O5	501.3589	[M-H]−	0.682	
1,3-Dihydroxy-23-oxocycloart-24-en-28-oic acid (=gardenolic acid B)d	23.6	C30H46O5	485.3278	[M-H]−	1.179	
Dikamaliartane Da or Fa	23.8	C30H46O4	469.2968	[M-H]−	1.314	
Gummiferartane 4e or 9e	24.3	C30H48O4	471.3483	[M-H]−	0.736	
1,3-Dihydroxy-23-oxocycloart-24-en-28-oic acid (=gardenolic acid B)d	24.5	C30H46O5	485.3283	[M-H]−	2.251	
Gummiferartane 4e or 9e	24.9	C30H48O4	471.3483	[M-H]−	0.608	
Dikamaliartane Da or Fa	25.3	C30H46O4	469.3328	[M-H]−	0.973	
Gummiferartane 4e or 9e	25.7	C30H48O4	471.3489	[M-H]−	1.966	
Dikamaliartane Da or Fa	25.9	C30H46O4	939.67328	[2M-H]−	1.423	
6,25-Dihydroxy-29-nor-3,4-secocycloarta-4(28),23-dien-3-oic acid; 25-Me ether, Me esteri or dihydroxy-methylenecycloartanoic acid	26.4	C31H50O4	485.3647	[M-H]−	2.177	
Dikamaliartane Da or Fa	26.9	C30H46O4	469.3331	[M-H]−	1.634	
Gummiferartane 4e or 9e	27.8	C30H48O4	483.3482	[M-H]−	0.407	
Notes.

All compounds assigned by comparison of accurate mass data (based on ppm#), and by interpretation of available MS/MS spectra.

a Reported to occur in Gardenia gummifera L.f. and G. lucida Roxb. (Kunert et al., 2009); the latter a synonym for G. resinifera Roth.

b Occurs in Gardenia carinata Wall. ex Roxb. (CCD, 2017).

c Occurs in Gardenia thailandica Tirveng. (CCD, 2017).

d Occurs in Gardenia jasminoides J.Ellis (CCD, 2017).

e Occurs in Gardenia gummifera (CCD, 2017).

f Constituent of the resin of Commiphora opobalsamum (L.) Engl. (CCD, 2017); synonym for Commiphora gileadensis (L.) C.Chr.

g Occurs in Gardenia obtusifolia Roxb. ex Hook.f. (CCD, 2017).

h D-xylo-form (guggultetrol 20) occurs in Commiphora mukul (Hook. ex Stocks) Engl. resin (CCD, 2017).

i Occurs in Antirhea acutata (DC.) Urb. (CCD, 2017); synonym for Stenostomum acutatum DC.

Also detected in the colleter exudate of Cheek 18345 by LC-MS were two compounds eluting at Rt 20.8 and 21.8 min that were both assigned the molecular formula C30H50O5 from their observed [M-H]− ions, which is that of gummiferartane 3, a cycloartane triterpenoid previously reported to occur in G. gummifera (CCD, 2017). Chemically related triterpenoids are gummiferartanes 4 and 9 that have the molecular formula C30H48O4 and also occur in G. gummifera (CCD, 2017); four compounds were assigned with this molecular formula in the colleter exudate, from their observed [M-H]− ions, eluting at Rt 24.3, 24.9, 25.7 and 27.8 min. Other cycloartane triterpenoids have previously been reported to occur in species of Gardenia (Kunert et al., 2009; CCD, 2017), with some of these in agreement with the molecular formulae of the triterpenoids detected in the colleter exudate of Cheek 18345, as indicated in Table 2.

Other compounds detected in the colleter exudate of Cheek 18345 included those that eluted at Rt 20.9 min with m/z 463.3281, and at Rt 21.6 min with m/z 391.3069, that were assigned the molecular formulae C24H48O8 and C20H42O4, respectively. These molecular formulae are those of 1,2,3,4-octadecanetetrol; 1-O-rhamnoside and 1,2,3,4-eicosanetetrol, respectively, which have been reported as components of the resin from Commiphora species in other studies, as indicated in Table 2.

Discussion

Employing chloroplast sequence data of tribe Pavetteae, largely based on De Block et al. (2015), placed the new Rubiaceae from Mt Gangan as sister to the rest of Clade II of that study, in which three genera, Leptactina, Dictyandra Hook.f. and Coleactina N.Hallé were traditionally maintained, although the two latter genera were recently subsumed into Leptactina s.l. Morphologically, the new Rubiaceae from Mt Gangan was consistent with these genera, especially Leptactina s.s. and Coleactina, yet showed significant character disjunctions, sufficient to support generic status. The new genus shares with the other members of Clade II large broad stipules and large calyx lobes, large flowers with pubescent corollas, intrusive placentas with numerous ovules and numerous small, angular seeds. However, morphological differences are marked (Table 1), notably the winged, glabrous pollen presenter (versus smooth and usually hairy in Leptactina s.l.), the absence of staminal connective appendages, the difference in ratio of calyx tube:lobe (calyx tube well-developed and conspicuous in the new taxon, versus absent or minute in Leptactina s.l except in Leptactina papalis (N.Hallé) De Block, formerly Coleactina papalis N.Hallé), the seeds being bicolored (however, the state of this character is unknown for several species of Leptactina and other Pavetteae), and the corolla tube having a narrow proximal part and a much wider and longer distal part (possibly unique in Pavetteae). The new Rubiaceae from Mt Gangan is atypical and differs from the standard state in all other genera of Pavetteae by having several-flowered axillary inflorescences (Fig. 4). This has been confirmed by observing the species during several seasons to ensure that the origin of the inflorescence is not terminal. However, some species of Pavetta, such as P. mayumbensis Bremek. also have such inflorescences, possibly by contraction of the short branches that bear terminal inflorescences in most species of that genus. The tribe is generally characterised by terminal inflorescences (De Block et al., 2015). However, in Clade II, the remarkable monotypic genus Coleactina from Gabon and the Republic of Congo, now included in Leptactina s.l., and the species Leptactina deblockiae Neuba & Sonké (Neuba, Malan & Kouadio, 2014) also have axillary inflorescences, albeit 1-flowered and not several-flowered. Finally, the copious and conspicuous bright red exudate from the apical bud of the new Rubiaceae from Mt Gangan appears to be unique in Pavetteae and probably Rubiaceae. Colleter exudates are common in Rubiaceae, but are usually inconspicuous. Conspicuous colleter-derived exudates are known in some genera in tribe Coffeeae, e.g., Coffea L., and in genera of other tribes, such as Gardenia J.Ellis. Although they are generally not reported in Pavetteae (Hallé, 1970; Bridson & Verdcourt, 1988; De Block et al., 2015), copious colleter exudate is present in the Madagascan Pavetteae genus Robbrechtia (De Block, 2003), and colleter exudate has also been observed in several other Pavetteae genera (P De Block, pers. comm., 2018). We have observed colleter exudates in some specimens of Leptactina (e.g., Fofana 188, Jacques-Felix 7422, both from Guinea, Leptactina senegambica Hook.f.; Goyder 6258, from Angola, Leptactina benguellensis (Benth. & Hook.f.) Good, all K!). As with all previously known Rubiaceae exudates except Gardenia (which is bright yellow, E Robbrecht, pers. comm., 2018), these are colourless or slightly yellow, and translucent, not bright red and opaque as in the new Rubiaceae from Mt Gangan.

Figure 4 K. gangan Cheek.

(A) habit, with indication of bullate leaf surface; (B) plants in situ on rock face (from photograph); (C) adaxial leaf indumentum around midrib; (D) abaxial leaf indumentum around midrib; (E) inner face of stipule at second node; (F) secretory colleter from E; (G) flower, post-anthetic; (H) peduncle and proximal cup of bracts with lobes (sheathing and concealing a smaller distal cup of bracts) below flower; (I) corolla from post-anthetic flower cut longitudinally and opened to display inner surface; (J) stigma; (K) transverse section of mature fruit, empty of seeds but showing placenta (in the left locule); (L) seed, hydrated, lateral view; (M) seed, dry, lateral view; (N) seed, dry, view from above. Scale bars: A, B = 5 cm; G, I, K = 1 cm; H = 5 mm; C, D, E, J = 2 mm; F, L, M, N = 1 mm. Drawn by Andrew Brown based on Cheek 18345.

The palynological differences between Kindia and Leptactina s.l. are extensive. All Leptactina s.l. have pollen type 2 (De Block & Robbrecht, 1998), i.e., the grains are circular to quadrangular in polar view, (3–)4-zonocolporate, with an apocolpial index of 0.39–0.68. In comparison, those of the new Rubiaceae from Mt Gangan are pollen type 1 (De Block & Robbrecht, 1998), since they are triangular in polar view (Fig. 2), 3-zonocolporate, with an apocolpial index of 0.125.

Possession of pollen type 1 by Cheek 18541A rather than pollen type 2, is consistent with its position as sister to Clade II since pollen type 1 ‘predominates in the whole of Rubiaceae and can be considered primitive’ (Robbrecht, 1988), that is, plesiomorphic. Pollen type 1 also occurs in Pavetteae Clades III and IV (De Block & Robbrecht, 1998; De Block et al., 2015). The four apertures of pollen type 2 are considered as derived (De Block & Robbrecht, 1998) and likely represent a synapomorphy for Leptactina s.l. in Clade II.

With the discovery, characterisation and placement of the new Rubiaceae of Mt Gangan as sister to Clade II, re-interpretation of the polarity of some characters in the rest of the clade is in order. Features of Leptactina papalis, previously interpreted as apomorphies for the genus Coleactina now appear to be plesiomorphic with regard to the newly discovered taxon. These are: the well-developed calyx tube, and the pair of involucral cups (cupular bracts) surrounding the ovary (Fig. 4H). Alternatively, these features may have evolved independently in both L. papalis and the new taxon. Additional potentially plesiomorphic characters for Clade II are the axillary inflorescences found in several Leptactina species including L. papalis and L. deblockiae (Neuba, Malan & Kouadio, 2014), and the new Rubiaceae of Mt Gangan. The newly discovered lineage, sister to the rest of Clade II, may represent an evolutionary relict, as it is only known from a single morphologically and molecularly isolated species, which is rare, with less than 100 individuals found in the wild. The unexpected discovery of this lineage from West Africa, sister to Leptactina s.l., which is most diverse in terms of species and morphology in Central Africa, e.g., in Gabon (Hallé, 1970) may also provide insights into the geographical origins of Clade II.

The unique habit of the new taxon within tribe Pavetteae may derive from adaptation to its unusual epilithic habitat: narrow fissures in vertical sandstone cliff faces (Figs. 1A and 1B). In this habitat, the well-developed aerial stems present in the rest of the tribe risk pulling the plants, by their mass, from the tiny fissures and pockets in which they are rooted. This circumstance appears to parallel the situation of Mussaenda epiphytica Cheek (tribe Mussaendeae, Rubiaceae; Cheek, 2009), a rare epiphytic species, similarly threatened with extinction (Onana & Cheek, 2011; Lachenaud et al., 2013), in a genus of shrubs and twining terrestrial climbers. Mussaenda epiphytica also appears to have mostly lost its ability to produce long stems, which was similarly conjectured to be disadvantageous in an epiphytic life form (Cheek, 2009). Several species of Leptactina are also subshrubs of nearly similar small stature to the new taxon, but these species have underground rootstocks and are terrestrial.

Plant exudates, including resins and gums, can occur as complex mixtures of different compound classes including carbohydrates, mono-, di- and tri-terpenoids (Rhourrhi-Frih et al., 2012). In this study, the colleter exudate of the new Rubiaceae from Mt Gangan was subjected to high resolution LC-MS/MS analysis for the first time to investigate the chemical composition and over 40 triterpenoids were detected including those assigned as the cycloartane class. These included those with the molecular formulae of dikamaliartanes A, D and F, or their isomers. The cycloartane triterpenoids, dikamaliartanes A–F have previously been subjected to antimicrobial assays using Staphylococcus aureus, Candida albicans and Mycobacteria but they did not reveal significant activity against these human pathogens (Kunert et al., 2009). Any potential role they may have against plant pathogens or as defence compounds requires further evaluation. Cycloartane triterpenoids are widely distributed in the plant kingdom and it has been suggested that cyclization of of (3S)-squalene 2,3-epoxide in higher plants occurs with formation of cycloartenol, which has been considered to have a role in sterol biosynthesis, analogous to that of lanosterol in animals and fungi (Boar & Roner, 1975). Furthermore, some plant triterpenoids, including those derived from cycloartane, have been suggested to have a function in cell membrane composition (Nes & Heftmann, 1981), thus any evolutionary role they may have in members of the new Rubiaceae from Mt Gangan would be of interest to explore in further studies. Many triterpenoids of plant origin have been of interest for their chemical diversity, biological activities and potential therapeutic applications (Hill & Connolly, 2017; Howes, 2018). The triterpenoids detected in the exudate in this study would be of further interest not only for their biological activities that might aid understanding of their rationale for synthesis by this species, but also for their potential uses by humanity, if this can be done in a way consistent with the conservation of this rare and threatened species.

Taxonomic Treatment

Kindia Cheek, gen nov.	

Type: Kindia gangan Cheek

Diagnosis: differs from Leptactina s.l. in a corolla tube with a slender proximal part and an abruptly much wider, longer distal part (not more or less cylindrical, or gradually widening); a glabrous, winged pollen-presenter (not hairy, non-winged); an epilithic habit (not terrestrial, growing in soil); a conspicuous opaque red colleter exudate (not translucent and colourless or slightly yellow); and type 1 pollen (not type 2) (De Block & Robbrecht, 1998).

Epilithic subshrub, lacking underground rootstock. Stems short, unbranched, erect or appressed to substrate, reiterating from base, completely sheathed in marcescent stipules, stem indumentum simple, short. Leaves opposite, petiolate, equal in shape and size at each node, each stem with 2–3 pairs of leaves held ± appressed to the vertical substrate; blades simple, entire; domatia absent; nervation pinnate; stipules broadly ovate, midline with a raised ridge; base of adaxial surface with a mixture of hairs and standard type colleters (Robbrecht, 1988) producing a vivid red exudate from the apical bud, conspicuous in dried specimens. Inflorescences axillary, opposite, in successive nodes, pedunculate-fasciculate, 1–4(–6)-flowered; bracts cupular, 2, sheathing, each with two large and two small lobes (Fig. 4H). Flowers 5-merous, homostylous. Ovary-hypanthium sessile, cylindric, with a ring of orange colleters inserted above the base, continuous with the calyx tube and about twice as long as broad; inside of the calyx tube with dense band of colleters at base, calyx lobes 5, oblong-elliptic, about as long as tube. Corolla nearly twice as long as calyx; tube infundibular-campanulate, exceeding calyx; outer surface densely sericeous, inner surface subglabrous apart from a dense band of hairs just above the base; corolla lobes 5, at anthesis elliptic-oblong, arching outwards (appearing broadly ovate when viewed from above Fig. 1C), with apiculus, post-anthesis drying elliptic-triangular (Fig. 4I), about one third as long as tube, aestivation contorted to the left in bud. Stamens adnate to the corolla tube, five, inserted midway up corolla tube, alternating with corolla lobes; anthers narrowly oblong, sessile, attached near base, apical appendage not developed. Ovary 2-celled, placentation axile; placentae intrusive, swollen, ovules numerous; style included, distal half hairy, basal part glabrous; pollen presenter (stylar head) dilated, outer surface glabrous, fluted-ridged, with two appressed stigmatic lobes at apex, apices tapering, acute, at same level as anthers. Fruit globose, ripening greenish-yellow or white, glossy, semi-translucent, outer surface hairy; pericarp succulent, thick, calyx persistent. Seeds numerous, truncated, 4–5-sided pyramidal (frustrums) glossy black; hilar area white, deeply excavated with a thickening inside; embryo occupying c. 5–10% of the seed volume, horizontal, cotyledons barely detectable.

Kindia gangan Cheek sp. nov.—Figs. 1 and 4	

Type. Republic of Guinea, Kindia Prefecture, Mt Gangan area, Kindia-Télimelé Rd, km 7, N of Mayon Khouré village, fr. 5 Feb. 2016, Cheek 18345 (holotype HNG!, isotypes BR!, K!, P!, US!).

Perennial epilithic subshrub, multi-stemmed from base. Stems very short, appressed to substrate or sometimes pendulous, not rooting at the nodes, woody, reiterating from base, completely sheathed in persistent dark brown stipules, 5–6(–35) cm long, each stem with 2–3 pairs of leaves held ± appressed to the substrate; internodes (2.5–)5 mm long, 5–7 mm diam.; indumentum composed of short white patent hairs, 0.1–0.2 mm long. Leaves opposite, equal in shape and size at each node; blade elliptic (-obovate), (7.5–)9.4–11.7 × (3.2–)4.2–6.6(–7) cm; apex obtuse to shortly acuminate, acumen 1–2 mm long; base acute, abruptly decurrent to the upper 2–5 mm of the petiole; upper blade surface bullate; indumentum white, subappressed, 0.1–0.3 mm long, 30% cover; midrib hairs 0.3–0.4 mm long, 80% cover; midrib c. 1 mm broad, yellow drying white; secondary nerves (7–)8–10(–11) on each side of the midrib; lower surface of blade with indumentum as upper, denser, c. 40% cover; midrib 1.2–1.3 mm wide, showing 3 distinct longitudinal areas; the central area raised, convex, 40% covered in hairs; the lateral areas flat, 90% covered in hairs; domatia absent; secondary nerves arising at c. 60° from the midrib, curving near the margin and looping towards the leaf apex and uniting with the nerve above (brochidodromous); tertiary nerves conspicuous, raised, white puberulent scalariform (5–)6–8 between each pair of secondary nerves; quaternary nerves apparent only in the tertiary cells (areolae) towards the margin, each tertiary cell with 8–12 bullae (not always visible in the pressed specimens). Petiole semi-circular in transverse section, 3–4 mm long at the distal-most node, elongating to 6–10(–14) mm long at the second and third node from the apex. Interpetiolar stipules broadly ovate 3–5.5 × 3–5 mm; apex acute or rounded to shortly acuminate; outer surface midline with a raised ridge; indumentum as upper surface of leaf blade; adaxial surface with colleters in line at the base, producing a vivid red exudate over the apical bud, conspicuous in dried specimens; colleters standard type (Robbrecht, 1988), orange, cylindric, 0.5–1.5 × 0.2 mm long, gradually tapering to a rounded apex, interspersed with bristly hairs 1–2 mm long at stipule base, otherwise hairs sparse, 0.2–0.4 mm long, 10–20% cover. Inflorescences axillary, opposite, and in successive nodes, pedunculate-fasciculate, 1–4(–6)-flowered. Peduncle 4–15 × 1.5–2.5 mm; indumentum as upper surface of leaf-blade; bracts cupular, 2, outer (proximal) bract sheathing and concealing the smaller inner (distal) bract, 3.5–4 × 5–7 mm, 4-lobed, with the large lobes (presumed of stipular origin) oblong-elliptic 4.5–6.5 × 2.5 mm and the short lobes (presumed of leaf origin) triangular, 1–2 × 2 mm. Ovary-hypanthium sessile (pedicel absent), partly concealed, and sunken in the axis below the insertion of the distal cupular bract (ovary locules extending below the junction of ovary with distal cupular bract), free part (that part which is not sunken into the axis) subcylindrical, 2 mm long, 4 mm in diameter at junction with calyx, hairs white, more or less patent, 0.5 mm long; ring of orange colleters 0.5–0.75 mm long, appressed, inserted about 1/3 up from base; calyx tube (3–)4–5(–10) × 4–5 mm at base, 5–6(–10) mm wide at apex; calyx lobes 5, oblong elliptic, 7–11 × 2–3(–4.5) mm, apex acute; indumentum on both surfaces 0.4–0.6(–1.1) mm long more or less patent, c. 50% cover on tube, 20–30% cover on lobes; inner surface also with a dense band of colleters at base, extending in lines a short distance up from the base of the calyx tube. Corolla white, infundibular-campanulate, 3–4.5 cm long pre-anthesis, at anthesis with lobes splayed, 4.2–4.3 cm wide at mouth including the lobes; outer surface of corolla densely pale brown sericeous, hairs 0.5 mm long, covering the surface; tube with two distinct sections, proximal and distal; proximal section slender, 6 × 2 mm, glabrous inside in proximal part, middle portion of the proximal tube with a densely puberulent band 1–2 mm long, hairs white 2 mm long forming a seal with the style; distal section of corolla tube abruptly wider, 2.2–2.6 × 1.4–1.6 cm, inner surface of distal section glabrous in proximal 2.2–2.4 cm, distalmost part of tube (at mouth) with thinly scattered hairs 0.1–0.2 mm long, 30–40% cover; lobes 5, glabrous inside, oblong-elliptic (appearing broadly ovate when viewed from above as in Fig. 1C), 9–12 × 6.5–9(–16) mm, then extending into a filiform appendage 3–4 mm long, apex acute, margins becoming involute post-anthesis. Stamens five, alternating with corolla lobes; anthers sessile, elliptic c. 5–6 × 1 mm, attached near the base and inserted 1.5 cm from corolla base. Disc bowl-shaped, 1 mm wide, 2 mm deep, glabrous, smooth. Ovary 2-celled, placentation axile; placentae intrusive, shield-shaped, 2 × 1.25 mm, 0.5 mm thick (including ovules); ovules 40–50 per locule, elliptic, 0.25 mm long; style included, 2.2 cm long, 1 mm diam. at base, proximal 9–9.5 mm glabrous, median 5–6 mm patent-hairy with hairs 0.3–0.5 mm long, distal 10.5–11 mm glabrous; pollen presenter (stylar head) dilated, with two appressed lobes 3 × 1–1.2 mm, outer surface fluted-ridged, apices tapering, acute. Fruit globose, 9–10 mm diam. sessile, ripening greenish-yellow or white, glossy, semi-translucent, outer surface with appressed white hairs 0.6–0.9 mm long; pericarp succulent, 2–3 mm thick, calyx persistent. Seeds numerous 30–50 per fruit, truncated, 4–5-sided, pyramid (frustum), 1.5–2 × 1.5–2 × 1.5 mm, the proximal (hilar end) white, the distal two-thirds glossy black; epidermis with finger-print surface pattern embryo minute, c. 0.3 mm long, cotyledons about 1/4 of length, not well demarcated.

Distribution

République de Guinée, Kindia Prefecture, northeastern boundary of Mt Gangan area, west of Kindia-Telimélé Rd (Fig. 5).

Figure 5 Map of the distribution of K. gangan.

The distribution of the species was mapped using SimpleMappr.

Ecology

The area of the Mt Gangan complex in which we found plants of Kindia consists of two parallel ranges of small sandstone table mountains separated by a narrow N–S valley that appears to be a geological fault. Bedding of the sandstone is horizontal. Uneven erosion on some slopes has resulted in the formation of frequent rock ledges, overhangs and caves. In contrast, other flanks of the mountains are sheer cliffs extending 100 m or more high and wide. It is on the cliff areas at 230–540 m a.s.l that K. gangan occurs as the only plant species present, usually as scattered individuals in colonies of (1–3–)7–15 plants, on the bare expanses of rock that are shaded for part of the day due to the orientation of the cliffs or to overhangs or due to a partial screen of trees in front of the rockfaces. Pitcairnia feliciana (Bromeliaceae), in contrast is found in fully exposed sites where there is, due to the rock bedding, a horizontal sill in which to root. These two species can grow within metres of each other if their cliff microhabitats occur in proximity. The rock formations create a variety of other microhabitats, including vertical fissures, caves, shaded, seasonally wet ledges, and are inhabited by sparse small trees, shrubs, subshrubs, perennial and annual herbs, many of which are narrow endemic rock specialists. We speculate that the seed of this species might be bat-dispersed because of the greenish yellow-white colour of the berries (less attractive to birds than fruits which are e.g., red or black) and the position of the plants high on cliff faces, where nothing but winged creatures could reach them, apart from those few plants at the base of the cliffs. However, fruit dispersal is not always effected since we found numerous old dried intact fruits holding live seeds on the plants at the type locality in February 2016. It is possible that the robust, large white flowers are pollinated by a small species of bat since in June and September we saw signs of damage to the inner surface of the corolla inconsistent with visits by small insects. The damage takes the form of brown spots on the inner surface of the corolla tube. Freshly opened flowers do not have these spots, nor do all flowers, only those few which show slight damage. The very broad, short corolla is not consistent with pollination by sphingid moths (which prefer long, slender-tubed flowers), but this cannot be ruled out.

Local names and uses

None are known. The local communities in the area when interviewed in November 2017, stated that they had no uses nor names for the plant (D Molmou & T Doré, pers. obs., 2017).

Etymology

The genus is named for the town and prefecture of Kindia, Guinea’s fourth city, and the species is named for Mt Gangan to its north, which holds the only known location for the species. Both names are derived as nouns in apposition.

Conservation status

Knowledge of K. gangan is based on 15 days of searching in sandstone rock outcrops around the Mt Gangan complex in 2016–2017 by teams each comprising 3–5 botanists, together with local community representatives. This area was previously visited by several excellent botanists in the colonial period, notably by Jacques-Félix in 1934–37. Only 86 mature plants of K. gangan were seen at seven sites at two locations (as defined by IUCN, 2012). The two locations are separated by 19 km. Within locations, the sites are separated by 150 m–1.5 km. The Extent of Occurrence and Area of Occupancy were calculated as 27.96 km2 and 20 km2 respectively (Bachman et al., 2011). At each site (1–7–)10–20 plants occur gregariously. Accordingly, since less than 250 mature individuals are known of this species, it is here assessed as Endangered under Criterion D1 of IUCN (2012). It is to be hoped that more plants will be found, enabling a lower assessment of the threat to this species. Currently, threats to the plants at the 2 known locations of this species are low. Quarrying of sandstone for building construction in nearby Kindia, Guinea’s fourth city occurs nearby, but fortunately one of the locations of K. gangan has no road access, so the known plants are not immediately threatened, while at the second location, plants are within reach of roads and so more threatened by future quarrying. It is to be hoped that further sites for the species will be found, lowering the extinction risk of the species. As a precautionary measure it is intended to feature the species in a poster campaign to raise public awareness, and to seedbank it in the newly created seed bank at the University of Gamal Abdel Nasser, Conakry and also at the Royal Botanic Gardens, Kew.

Additional specimens examined

Republic of Guinea, Kindia Prefecture, Mt Gangan area, Mt Gnonkaoneh, NE of Mayon Khoure village which is W of Kindia-Télimelé rd., fl. 19 June 2016, Cheek 18529 (HNG!, K!); Mt Khonondeh, NW of Mayon Khoure village which is W of Kindia to Télimelé rd., fl. 20 June 2017, Cheek 18545 (HNG!, K!). Mt Gnonkaoneh, NE of Mayon Khoure village, fl. 30 Sept. 2016, Cheek 18602 (HNG!, K!); near Kalakouré village, Kindia-Télimelé rd, fr. 1 Nov. 2017, Doré 136 (HNG!, K!); Sougorunyah near Fritaqui village, fr. 6 Nov. 2017, Molmou 1669 (HNG!, K!); Kebe Figuia near Fritaqui village, fr. 6 Nov. 2017, sight observation by Doré and Molmou. Additional observation (photo record): Mt Khonondeh, NW of Mayon Khoure village which is W of Kindia to Télimelé rd., fl. 20 June 2017, Cheek 18541A.

Conclusions

Kindia, an endangered subshrub, restricted to bare, vertical rock faces of sandstone is described and placed in Clade II of tribe Pavetteae as sister to Leptactina s.l. based on chloroplast sequence data. The only known species, K. gangan, is distinguished from the species of Leptactina s.l. by a combination of characters: an epilithic habit; several-flowered axillary inflorescences; distinct calyx tube as long as the lobes; a infundibular-campanulate corolla tube with narrow proximal section widening abruptly to the distal section; presence of a dense hair band near base of the corolla tube; anthers and style deeply included, reaching about mid-height of the corolla tube; anthers lacking connective appendages and with sub-basal insertion; pollen type 1; pollen presenter winged and glabrous; orange colleters, which encircle the calyx-hypanthium, occur at base and inside the calyx and stipules and produce vivid red exudate. High resolution LC-MS/MS analysis revealed over 40 triterpenoid compounds in the colleter exudate, including those assigned to the cycloartane class. Triterpenoids are of interest for their diverse chemical structures, varied biological activities, and potential therapeutic value.

Supplemental Information

Data S1 Concatenated alignment of the chloroplast sequence data (rps16 and trnT-F)

Click here for additional data file.

Figure S1 Majority consensus multiple-locus BI cladogram with the associated PP values and the BS values of the multiple-locus ML tree

Only PP above 0.80 and BS values above 75% are shown. Nodes with PP < 0.5 support have been collapsed. Inset tree shows the branch lengths.

Click here for additional data file.

Professor Basile Camara, former Director General of the Université Gamal Abdel Nasser de Conakry-Herbier National de Guinée, is thanked for arranging permits and for his long term support and collaboration. Janis Shillito is thanked for typing the manuscript. Charlie Gore assisted with scanning electron microscopy. The authors would like to thank Dr Geoffrey C. Kite, Royal Botanic Gardens, Kew, for acquiring the LC-MS data. Three reviewers, Dr Petra De Block, Dr Olivier Lachenaud and Prof. Elmar Robbrecht are thanked for constructive comments on earlier drafts of the paper.

Appendix

Sampled plants and DNA sequences. For each plant the provenance, followed by collector and collector number, herbarium for deposition of voucher specimen (in parentheses), and GenBank accession numbers for rps16 and trnT-F. FTEA: Flora of tropical East Africa. Abbreviation ‘s.n.’ indicates no collection number. The newly generated sequences are in bold.

Tribe Alberteae: Razafimandimbisonia humblotii (Drake) Kainul. & B.Bremer—Madagascar, Tosh et al. 263 (BR), KM592238, KM592145.

Tribe Coffeeae: Tricalysia semidecidua Bridson—Zambia, Dessein et al. 1093 (BR), KM592279, KM592185.

Tribe Ixoreae: Ixora sp.—Thailand, Sudde 1487 (K), KM592208, KM592115.

Tribe Gardenieae: Euclinia longiflora Salisb.—Africa (country unknown), Van Caekenberghe 348 (BR), KM592203, KM592110.

Gardenia rutenbergiana (Baill. ex Vatke) J.-F.Leroy—Madagascar, Groeninckx et al. 24 (BR), KM592204, KM592111.

Oxyanthus troupinii Bridson—Burundi, Niyongabo 115 (BR), KM592219, KM592126.

Tribe Mussaendeae: Pseudomussaenda flava Verdc.—Africa (country unknown), Van Caekenberghe 60 (BR), KM592217, KM592124.

Tribe Pavetteae: Cladoceras subcapitatum (K.Schum. & K.Krause) Bremek.—Tanzania, Luke et al. 8351 (UPS), AM117290, KM592094.

Coptosperma bernierianum (Baill.) De Block—Madagascar, Schatz et al. 3764 (MO), KJ815340, KJ815589; C. borbonicum (Hend. & Andr.Hend.) De Block—Comores, De Block 1389 (BR), KM592189, KM592096; C. borbonicum (Hend. & Andr.Hend.) De Block—Réunion, Kainulainen 189 (S), KJ815342, KJ815591; C. borbonicum (Hend. & Andr.Hend.) De Block—Unknown, Kroger et al. 56 (S), KJ815341, KJ815590; C. cymosum (Willd. ex Schult.) De Block—Mauritius, Razafimandimbison et al. 843 (S), KJ815343, KJ815592; C. graveolens (S.Moore) Degreef—Kenya, Mwachala 3711 (BR), KM592200, KM592107; C. humblotii (Drake) De Block—Madagascar, Bremer et al. 5167 (S), KJ815345, KJ815594; C. littorale (Hiern) Degreef—Mozambique, Luke et al. 9954 (UPS), KM592190, KM592097; C. madagascariense (Baill.) De Block—Madagascar, De Block et al. 2238 (BR), KM592191, KM592098; C. madagascariense (Baill.) De Block—Madagascar, Razafimandimbison 527 (UPS), KM592191, KM592098; C. mitochondrioides Mouly & De Block—Madagascar, Bremer et al. 5127 (S), KJ815348, KJ815597; C. nigrescens Hook.f.—Madagascar, De Block et al. 535 (BR), KM592192, KM592099; C. nigrescens Hook.f.—Kenya, Luke & Luke 9030 (UPS), KM592193, KM592100; C. peteri (Bridson) Degreef—Tanzania, Lovett & Congdon 2991 (BR), KM592201, KM592108; C. supra-axillare (Hemsl.) Degreef—Madagascar, De Block et al. 1321 (BR), KM592194, KM592101; C. sp. nov. A—Madagascar, De Block et al. 720 (BR), KM592199, KM592106; C. sp. nov. B—Madagascar, De Block et al. 796 (BR), KM592195, KM592102; C. sp. nov. C—Madagascar, De Block et al. 1355 (BR), KM592196, KM592103; C. sp. nov. D—Madagascar, De Block et al. 704 (BR), KM592197, KM592104; C. sp. nov. E—Madagascar, De Block et al. 733 (BR), KM592198, KM592105.

Homollea longiflora Arènes—Madagascar, De Block et al. 767 (BR), KM592205, KM592112; H. perrieri Arènes—Madagascar, Morat 4700 (TAN), KM592206, KM592113.

Kindia gangan Cheek—Republic of Guinea, Cheek 18345 (K), MG708505, MG708506.

Leptactina arborescens (Welw. ex Benth. & Hook.f.) De Block—Ghana, Schmidt et al. 1683 (MO), KM592202, KM592109.; L. benguelensis (Welw. ex Benth. & Hook.f.) R.D.Good—Zambia, Dessein et al. 1142 (BR), KM592209, KM592116; L. delagoensis K.Schum.—Tanzania, Luke & Kibure 9744 (UPS), KM592210, KM592117; L. epinyctios Bullock ex Verdc.—Zambia, Dessein et al. 1348 (BR), KM592211, KM592118; L. involucrata Hook.f.—Cameroon, Davis 3028 (K), KM592212, KM592119; L. leopoldi-secundi Büttner—Republic of Congo, Champluvier 5248 (BR), KM592213, KM592120; L. mannii Hook.f.—Gabon, Dessein et al. 2518 (BR), KM592214, KM592121; L. papalis (N.Hallé) De Block—Gabon, Dessein et al. 2355 (BR), KM592188, KM592095; L. papyrophloea Verdc.—Tanzania, Luke & Kibure 9838 (UPS), KM592215, KM592122; L. pynaertii De Wild.—Republic of the Congo, Champluvier s.n. (BR), KM592216, KM592123.

Nichallea soyauxii (Hiern) Bridson—Cameroon, Dessein et al. 1402 (BR), KM592218, KM592125.

Paracephaelis cinerea (A.Rich. ex DC.) De Block—Madagascar, De Block et al. 2193 (BR), KM592220, KM592127; P. cinerea (A.Rich. ex DC.) De Block—Madagascar, Bremer et al. 5122 (S), KJ815372, KJ815619; P. saxatilis (Scott-Elliot) De Block—Madagascar, De Block et al. 2401 (BR), KM592221, KM592128; P. saxatilis (Scott-Elliot) De Block—Madagascar, Razafimandimbison & Kroger 937 (S), KJ815374, KJ815622; P. sericea (Arènes) De Block, Madagascar, De Block et al. 849 (BR), KM592207, KM592114; P. tiliacea Baill.—Madagascar, Groeninckx et al. 113 (BR), KM592222, KM592129; P. trichantha (Baker) De Block—Aldabra (Seychelles), Friedmann 833385 (UPS), KJ815376, KJ815624; P. sp.—Madagascar, De Block 1174 (BR), AM117331, KJ815620.

Pavetta abyssinica Fresen.—Africa (unknown country), De Block 6 (BR), FM204726, FM207133; P. agrostiphylla Bremek.—Sri Lanka, Bremer B. & K. 936 (UPS), KM592223, KM592130; P. batesiana Bremek.—Gabon, Dessein et al. 2071 (BR), KM592224, KM592131; P. hymenophylla Bremek.—Tanzania, Luke et al. 9101 (UPS), KM592225, KM592132; P. indica L.—Sri Lanka, Andreasen 202 (UPS), KM592226, KM592133; P. sansibarica K.Schum.—Kenya, Luke et al. 8326 (UPS), KM592227, KM592134; P. schumanniana F.Hoffm. ex K.Schum.—Zambia, Dessein et al. 911 (BR), KM592228, KM592135; P. stenosepala K.Schum.—Kenya, Luke et al. 8318 (UPS), KM592233, KM592140; P. suffruticosa K.Schum.—Cameroon, Lachenaud et al. 838 (BR), KM592231, KM592138; P. tarennoides S.Moore—Kenya, Luke et al. 8325 (UPS), KM592234, KM592141; P. ternifolia Hiern—Burundi, Ntore 19 (BR), KM592235, KM592142; P. tetramera (Hiern) Bremek—Gabon, Van de Weghe 163 (BR), KM592236, KM592143; P. vaga S.T.Reynolds—Australia, Harwood 1290 (DNA), KM592237, KM592144; P. sp. A of FTEA Bridson—Tanzania, Luke et al. 9134 (UPS), KM592232, KM592139; P. sp. B—Vietnam, Davis et al. 4082 (K), KM592229, KM592136; P. sp. C—Asia (country unknown), Van Caekenberghe 199 (BR), KM592230, KM592137.

Robbrechtia grandifolia De Block—Madagascar, Kårehed 311 (UPS), KM592239, KM592146; R. milleri De Block—Madagascar, Bremer et al. 5295 (S), KM592240, KM592147.

Rutidea decorticata Hiern—Cameroon, Maurin 14 (K), KM592241, KM592148; R. dupuisii De Wild.—Gabon, Dessein et al. 1802 (BR), KM592242, KM592149; R. ferruginea Hiern—Cameroon, Dessein et al. 1674 (BR), KM592242, KM592150; R. fuscescens Hiern—Tanzania, Luke et al. 9124 (UPS), KM592244, KM592151; R. membranacea Hiern—Liberia, Adam 21433 (UPS), KM592245, KM592152; R. olenotricha Hiern—Ghana, Schmidt et al. 1731 (MO), KM592246, KM592153; R. parviflora DC.—Liberia, Adam 20156 (UPS), KM592248, KM592154; R. seretii De Wild.—Cameroon, Gereau 5588 (UPS), KM592249, KM592155.

Schizenterospermum grevei Homolle ex Arènes—Madagascar, De Block et al. 2167 (BR), KM592250, KM592156; S. rotundifolia Homolle ex Arènes—Madagascar, De Block et al. 771 (BR), KM592251, KM592157.

Tarenna alleizettei (Dubard & Dop) De Block—Madagascar, De Block et al. 1883 (BR), KM592272, KM592178; T. alleizettei (Dubard & Dop) De Block—Madagascar, Kårehed 313A (UPS), KJ815382, KJ815630; T. alpestris (Wight) N.P.Balakr.—India, De Block 1474 (BR), KM592252, KM592158; T. asiatica (L.) Kuntze ex K.Schum.—India, Auroville 998 (SBT), KM592253, KM592159; T. bipindensis (K.Schum.) Bremek., Liberia, Jongkind 8495 (BR), KM592255, KM592161; T. capuroniana De Block—Madagascar, De Block et al. 937 (BR), KM592273, KM592179; T. capuroniana De Block—Madagascar, Bremer et al. 5041 (S), KJ815386, KJ815634; T. depauperata Hutch.—China, Chow & Wan 79063 (UPS), KM592256, KM592162; T. flava Alston—Sri Lanka, Klackenberg 440 (S), KM592257, KM592163; T. fuscoflava (K.Schum.) S.Moore—Ghana, Schmidt et al. 2099 (MO), KM592258, KM592164; T. gracilipes (Hayata) Ohwi—Japan, Van Caekenberghe 149 (BR), KM592259, KM592165; T. grevei (Drake) Homolle—Madagascar, De Block et al. 959 (BR), KM592274, KM592180; T. jolinonii N.Hallé—Gabon, Champluvier 6098 (BR), KM592260, KM592166; T. lasiorachis (K.Schum. & K.Krause) Bremek.—Gabon, Wieringa 4432 (WAG), KM592261, KM592167; T. leioloba (Guillaumin) S.Moore—New Caledonia, Mouly 174 (P), KM592262, KM592168; T. microcarpa (Guillaumin) Jérémie—New Caledonia, Mouly 297 (P), KM592263, KM592169; T. nitidula (Benth.) Hiern—Liberia, Jongkind 8000 (BR), KM592264, KM592170; T. pallidula Hiern—Gabon, Dessein et al. 2215 (BR), KM592265, KM592171; T. pembensis J.E.Burrows—Mozambique, Luke et al. 10136 (UPS), KM592266, KM592172; T. precidantenna N.Hallé—Gabon, Dessein et al. 2360 (BR), KM592267, KM592173; T. rhypalostigma (Schltr.) Bremek.—New Caledonia, Mouly 182 (P), KM592268, KM592174; T. roseicosta Bridson—Tanzania, Luke et al. 9170 (UPS), KM592269, KM592175; T. sambucina (G.Forst.) T.Durand ex Drake—New Caledonia, Mouly et al. 364 (P), KM592270, KM592176; T. spiranthera (Drake) Homolle—Madagascar, De Block et al. 946 (BR), KM592275, KM592181; T. thouarsiana (Drake) Homolle—Madagascar, De Block et al. 655 (BR), KM592276, KM592182; T. uniflora (Drake) Homolle—Madagascar, Bremer et al. 5230 (S), KM592277, KM592183; T. vignei Hutch. & Dalziel—Republic of Guinea, Jongkind 8126 (BR), KM592271, KM592177.

Tennantia sennii (Chiov.) Verdc. & Bridson—Kenya, Luke et al. 8357 (UPS), KM592278, KM592184.

Tribe Vanguerieae: Vangueria madagascariensis J.F.Gmel.—Africa (country unknown), Delprete 7383 (NY), EU821636.

Additional Information and Declarations

Competing Interests

Author Contributions

Field Study Permissions

DNA Deposition

Data Availability

New Species Registration

The authors declare there are no competing interests.

Martin Cheek, Melanie-Jayne R. Howes and Isabel Larridon conceived and designed the experiments, performed the experiments, analyzed the data, contributed reagents/materials/analysis tools, prepared figures and/or tables, authored or reviewed drafts of the paper, approved the final draft.

Sékou Magassouba, Tokpa Doré, Saïdou Doumbouya, Denise Molmou, Aurélie Grall and Charlotte Couch performed the experiments, contributed reagents/materials/analysis tools, authored or reviewed drafts of the paper, approved the final draft.

The following information was supplied relating to field study approvals (i.e., approving body and any reference numbers):

Permits to export the studied specimens were issued by the Ministère de l’Environnement et des Eaux et Forêts of the Republic of Guinea, Certificat d’Origine no 0000344 (date 21 June 2016) and no 0000399 (dated 28 October 2016).

The following information was supplied regarding the deposition of DNA sequences:

The newly generated sequence data are not yet publicly available, but have been deposited to GenBank, and are available as part of the Data S1 (sequence alignment used for the analyses).

The following information was supplied regarding data availability:

GenBank accession numbers included in Appendix 1 and in Data S1 (sequence alignment used for the analyses).

The following information was supplied regarding the registration of a newly described species:

Genus name: Kindia, LSID: 77177782-1.

Species name: Kindia gangan, LSID: 77177783-1.

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
