# Peer review of "Kindia (Pavetteae, Rubiaceae), a new cliff-dwelling genus with chemically profiled colleter exudate from Mt Gangan, Republic of Guinea"

_PeerJ, doi:10.7717/peerj.4666_

## Round 0.1 · original submission · Major Revisions

As can be seen from the Reviews and what is my own Impression your work makes an important contribution to the field. However, the manuscript can be significantly improved by the suggested changes and corrections. The writing should be more focussed and repetitions should be omitted. In some cases figures and descriptions seem to deviate as the morphology of the new species is concerned. Please go through the suggested points and integrate the improvements into the manuscript commenting on each of them in a separate letter, also on the points you decide not to consider.

·

Basic reporting

This paper forms a coherent publication; it is clear and generally well-written (but see General comments for some suggested improvements in style/terminology). The structure of the manuscript is accurate and the introduction provides an appropriate context. The illustrations are relevant and appropriately labelled (though I would recommend to include the exact posterior probability values in Fig. 3). The relevant literature is cited.

Experimental design

The research presented here is certainly original, and falls within the scope of the journal. The investigation has been appropriately conducted and the methods are accurately described.

Validity of the findings

The authors' conclusion that their plant represents a new genus and species is well-supported, both by molecular and morphological data. However, some modifications are needed in the genus and species descriptions, and in the discussion of generic characters (see General comments). I will not comment on the chemistry, which is outside my area of expertise.

Additional comments

This paper is of great interest for botany, and may be accepted for publication, but significant revision is required, particularly concerning the following points.

The corolla tube of the new genus is described as "dimorphic" (line 29 & Table 1) but this term is clearly incorrect, as it implies two different types of flowers. The correct term for a corolla tube with a narrow basal portion and an abruptly widening upper one is "campanulate" (meaning literally bell-shaped, as e.g. in a Campanula flower). The inflorescences of the new genus are repeatedly described as "many-flowered", but in fact there are only 1-4(-6) flowers per inflorescence, which is not that many.

In the introduction, the authors report that "botanical exploration, discovery and publication of new species [in Guinea] appeared to have nearly stopped after Independence in 1960. Those few species that were published in the period 1960–2010 were based on specimens collected in the French Colonial period, e.g. Phyllanthus felicis Jean F.Brunel (1987) and Clerodendrum sylviae J.-G.Adam (1974)." This is not entirely true, first because Guinea became independent in 1958 (not 1960), and second because Stanislaw Lisowski made important collections in the country from the late 1970s to the early 1990s, based on which several new species were described - for example Pseudoprosopis bampsiana Lisowski, Mikaniopsis camarae Lisowski, or Bacopa lisowskiana Mielcarek.

The authors list Dissotis controversa (A. Chev. & Jacq.-Fél.) Jacq.-Fél. among the endemics of Mt Gangan, but this species has been reduced to varietal rank, as Melastomastrum theifolium (G.Don) A.Fern. & R.Fern. var. controversum (A. Chev. & Jacq.-Fél.) Jacq.-Fél. (Bull. Mus. Natl. Hist. Nat., Ser. 3, Bot. 17: 71, 1975). The other endemics listed are good species.

The genus Karima is listed among recent discoveries in Guinea, but is not a very pertinent example here, since it is relatively widespread outside the country, and its only species was discovered more than a century ago (though it was misplaced in the large genus Croton).

The discussion of diagnostic characters between Kindia and Leptactina (lines 307-318, and Table 1) needs some modifications. The pollen, and the corolla tube shape and internal indumentum, are clear differences between the two genera. The colleter exsudate is another difference, which is mentioned in the text but surprisingly omitted from Table 1 - please include it there! No differences in the seeds are claimed, but there appear to be some: the seeds of Kindia are bicoloured, while those of Leptactina, as far as I know, are all black. On the other hand, some characters are more variable in Leptactina than implied by the authors: for example, the style arms of Leptactina are not always divergent at anthesis (see the excellent illustration of L. pynaertii in Flore du Gabon vol. 17); the stamen insertion is sub-basal in L. arborescens, as in Kindia; the calyx tube of L. papalis is well-developed; the staminal connective appendage is, as far as I remember, not always present in Leptactina (check in particular L. arborescens) and I am not sure the pollen presenter of Leptactina is always hairy.

It is also claimed (lines 313-315) that "The new Rubiaceae from Mt Gangan differs from all other genera of Pavetteae by having many-flowered axillary inflorescences", but in fact, some species of Pavetta do have many-flowered axillary inflorescences - for example P. mayumbensis Bremek.

The generic and specific description are partly redundant; to limit this, either the former can be shortened, or the two descriptions combined into one (which is relatively common practice in the case of monospecific genera).

Some discrepancies between the generic/specific description and the illlustrations (Fig. 1 & 4) have been noted, and it is especially important to check and correct them:
- in the generic description, the corolla lobes are described as "elliptic-triangular" but Fig. 1 shows them to be broadly ovate and apiculate at apex.
- the bracts are described as "cupular, 2, sheathing, with two large and two small lobes" in the generic description, and "cupular, 2, outer (proximal) sheathing the smaller inner (distal), 3.5–4 by 5–7 mm, large lobes oblong-elliptic 4.5–6.5 by 2.5 mm, short lobes triangular 1–2 by 2 mm" in the species description. Actually, Fig. 4H shows four bracts in a single row, fused together at base.
- in the species description the corolla is described as "white, cylindrical, 4–4.5 cm long, 2–2.3 cm wide at mouth" but on Fig. 4G it appears to be significantly narrower, judging from the scale bar (it is also unclear whether the size of 4–4.5 cm long refers to the whole corolla, or to the tube only).
- the inner surface of the corolla is described as "glabrous in proximal 2.2–2.4 cm, distal part of tube with thinly scattered hairs 0.1–0.2 mm long, 30–40 % cover". Actually, Fig. 4I shows a ring of hairs near the base, and no indumentum in the distal part of the tube - either the description or the illustration needs to be corrected.

Some parts of the specific description are not structured in a very logical way, particularly as far as the leaves are concerned: it is common practice to describe first the petiole, then the leaf blade (shape, size, base, apex, upper surface, lower surface) and then the venation (midrib, secondary veins, tertiary veins).

The disk is described as "adnate to base of corolla tube", but I am doubtful about this - it would be unique in Rubiaceae, where the disk is normally free from the corolla tube, though included in it.

Other comments and corrections of lesser importance have been made in the text of the paper.

·

Basic reporting

See attached PDF

Experimental design

See attached PDF

Validity of the findings

See attached PDF

Additional comments

See attached PDF

·

Basic reporting

The manuscript is well-written by native English speakers. There are no typographical errors although there are some small mistakes the authors missed during the rereading of the msc. These are indicated in the annotated msc.
The authors have good knowledge of the study site and the new species they describe. They cite most of the appropriate references. One reference is missing from the list and one reference listed is not cited in the text, though. See annotated msc for details.
Some additions as to the origin of the materials used in the molecular study have been added. These data are available on herbarium websistes such as Tropicos.

The structure of the article is clear and the figures and tables are all needed.
The manuscript certainly deserves publication in PeerJ.

The structure of the manuscript is good. But at some points the msc is rather exhaustive, notably in the introduction and the Material and methods section
-Botanical exploration and new species discovery in Guinea: all the recently described new species of the region are given in full and all the references are cited (all references of Kew authors). I would suggest this part is shortened. Three examples of new species is surely more than enough.
-Mt Gangan: a Tropical Important Plant Area: a detailed description of the study site is given. Some of this text is copied word for word in the taxonomic treatment (Ecology). Changing this somewhat would be better.
-Material and methods: is it really necessary to cite a reference for the way a specimen is collected in the field?

I have two main concerns about this msc.
1. the species description of Kindia gangan: This description is very, very exhaustive. Many details are unnecessary because most botanists would never measure/describe certain of the organs the authors measure/describe, e.g. the width of midrib on upper and lower leaf surface, the pubescence cover of the three parts of the midrib on the upper leaf surface, the width of the peduncle, the diameter of the internodes, the shape of the apex of the colleters, the width and depth of the disc, etc. These details are only useful if they are also known for other species that could possibly be confused with K. gangan or if they can be used to differentiate the new genus from Leptactina. But these details are unknown for other species. They make the description difficult to read and interpret. The authors also unnecessarily describe certain characters,
e.g. subshrub, multistemmed from base (but this is the definition of a shrub)
e.g. secondary nerves curving near the margin and looping towards the leaf apex and uniting with the nerve above (but this is the definition of brochidodromous)
To bring structure the authors use "." and ",". They only rarely use ";" but this could really help the reader to separate the different parts of the description. Removing the unnecessary details and restructuring the description should be mandatory before acceptance of the article for publication.
There are a few mistakes in the description: stems completely sheathed in persistent dark brown stipules, 5–6(–35) cm long. This length is certainly wrong. The authors also cite the size of the stamens c. 1.5 by 0.1 mm; according to the line drawing, this is wrong and should rather be 0.6 x 0.1 cm. Please check.
For me, the length and width of the corolla tube is confusing in the description: The authors write: Corolla white, cylindrical, 4–4.5 cm long, 2–2.3 cm wide at mouth; with two distinct sections; proximal section slender, 6 by 2 mm; distal section of corolla tube abruptly wider, 2.6 by 1.4 cm. What does 2.6 by 1.4 cm stand for? Is 2.6 the length of the distal section? In that case, 2.6 +0.6 is not 4-4.5 cm. What does 1.4 cm stand for? Is this the width of the distal part? This is different from 2-2.3 cm wide given above.

2. The comparison with the genus Leptactina is not always complete/correct. This is apparent in Table 1 and in the discussion.
Discussion
-“the difference in ratio of calyx tube:lobe (calyx tube well-developed and conspicuous in the new taxon, versus absent or minute in Leptactina s.l.).” There is always a calyx tube in Leptactina. And the calyx tube is very well-developed and much longer than the lobes in L. papalis.
-The authors call the inflorescence of K. gangan many-flowered in this part of the discussion. However 1-4(-6) flowers per inflorescence is pauciflorous in the Pavetteae which can have up to 250 flowers in many-flowered inflorescences.
-Copious colleter exudate is present in the Madagascan Pavetteae genus Robbrechtia (De Block 2003). It is also present in several other Pavetteae genera but this has not been published.
Table 1
-The pollen in Leptactina are (3-)4 colporate, not just 4-colporate
- The corolla tube in Leptactina is narrow but widens somewhat at the throath in the region where the stamens are included.
- De Block & Robbrecht (1989) cite AI as 0.39-0.68; where does your value come from?
- I would suggest splitting up the character of the position of stigma and anthers. In Leptactina the stigma is usually found at the level of the throath but rarely much deeper in the corolla tube, e.g. in L. epinyctios and see other examples in Puff et al. (1996). The stamens are always found in the upper part of the corolla tube, often with the tips excluded.
- Corolla tube length: breadth ratio
The authors cite (15-)20-25:1 for Leptactina. This is true in most species, but some species have short corolla tubes, e.g. L. arborescens. In those species, the ratio could be as low as 5:1 (see e.g. Hallé 1970)
As for Kindia gangan, the authors cite 3:1. According to the middle photo below in the habit photographs the ratio should be 2:1 rather than 3:1. And this is also the case according to the description: tube 4–4.5 cm long, 2–2.3 cm wide at mouth.
-The outside (abaxial surface) of the stigma lobes in Leptactina is usually, but not always, hairy
-Please cite references for the characters of Leptactina if not observed by yourselves.

The authors do not mention the colouring on the inside of the corolla tube and the base of the corolla lobes (also on the inside), which is spotted with reddish-brown spots. This character is certainly correlated with the shape of the corolla tube but deserves mentioning in the species description. Also, all Leptactina species have an all-white corolla (inside and outside).
The authors mention that the calyx is persistent on the fruit. However, the fruit picture seems to suggest otherwise. A non-persistent calyx on the fruit would also distinguish the new genus from Leptactina.
The authors never mention the fact that a subshrub habit is rather common in Leptactina. While there are no cliff-dwelling species (as far as known), it would be interesting to discuss the relative high occurrence of dwarfy shrubs in this genus (in the whole clade II). Examples are L. angolensis, L. pretrophylax, L. benguelensis, etc.
-Did the authors notice any flower scent while collecting the plants?

References:
- De Block P. (2003) Robbrechtia, a new Rubiaceae genus from Madagascar. Syst. Bot. 28: 145-156.
- Puff C., Robbrecht E., Buchner R. & De Block P. (1996) Survey of secondary pollen presentation in the Rubiaceae. Proceedings of the second international Rubiaceae conference. Op. Bot. Belg. 7: 369 402.

Experimental design

The experimental design for the description of a new species is not exhaustive. There are no great experiments to be done. The methods of the authors and the data presented by them is adequate. Their ecological knowledge of the species is a good bonus. Most new species are described from herbarium material. Here, a detailed study gives good information about habitat and population size.

The chemical profiling of the colleter exudate is superfluous in the context of describing a new species. However, not much is known about the colleter exudate of Rubiaceae and it would be good for this data to be published, so I would not object to it being included in the paper.

Validity of the findings

The findings are valid. This is certainly a new species from an interesting area in Western Africa. I think the description of this new species in a new genus is valid. Publishing this new taxon will bring attention to this region and help its conservation.

---

## Round 0.2 · Minor Revisions

I think that the referees did an excellent job to help and make the manuscript more concise and complete. Please respond to their remaining minor issues

·

Basic reporting

Nothing to add from my previous review. The paper is clear, generally well-written, and forms a coherent publication. The structure of the manuscript is accurate, and so are the illustrations. The relevant literature has been cited.

Experimental design

The research presented here is certainly original, and falls within the scope of the journal. The investigation has been appropriately conducted and the methods are accurately described.

Validity of the findings

The authors' conclusion that their plant represents a new genus and species is well-supported, both by molecular and morphological data.

Additional comments

As noted in my previous review, this paper is of great interest for botany, and definitely worth publishing. The revised version of the paper has been significantly improved, and most of the reviewers' comments have been taken into account by the authors. However, some points remain in need of revision. The most important of them are listed below. Other comments of lesser importance have been made directly in the paper.

Introduction:

- Dissotis controversa (A. Chev. & Jacq.-Fél.) Jacq.-Fél., cited among the endemics of Mt Gangan, is endemic indeed but no longer accepted as a species. Jacques-Félix (Bull. Mus. Natl. Hist. Nat., Ser. 3, Bot. 17: 71, 1975) transferred this taxon to Melastomastrum and reduced it to varietal rank, as M. theifolium var. controversum (A.Chev. & Jacq.-Fél.) Jacq.-Fél. He was the expert on African Melastomataceae, and as far as I know, no recent taxonomic publication has challenged his view, which is accepted e.g. in the African Plants Database. So, the acceptance of the name D. controversa by The Plant List seems to be due to an oversight.

- Among the differences between Kindia and Sabicea, the campanulate corolla tube and bicoloured seeds should also be mentioned (Sabicea spp. have uniformly brown seeds and a much narrower, almost cylindrical corolla tube).


Discussion:

- As already stated in my first review, the first paragraph of the discussion (and also the generic diagnosis and Table 1) should stress an additional difference between Kindia and Leptactina: the seeds are bicoloured in the former vs. all-black in the latter. Even though the authors note that "There is no survey or report of seed colour in Leptactina s.l., and we have not been able to survey all species for that", I have been able to survey the material in the Meise Botanic Garden, and I can confirm that all species of Leptactina with adequate fruiting material for study do have all-black seeds. (Species studied: L. arborescens, benguellensis, congolana, delagoensis, densiflora, hexamera, involucrata, leopoldi-secundi, laurentiana, liebrechtsiana, mannii, platyphylla, pynaertii, senegambica. Species not studied for this character due to the absence of mature fruits: L. angolensis, deblockiae, epinyctios, oxyloba, papalis, papyrophloea, rheophytica). If a reference is needed for this statement this may be cited as "Lachenaud, pers. comm.". The bicoloured seeds of Kindia in fact seem to be unique in Rubiaceae, at least in Africa.

- The section between lines 322 and 332 should put more emphasis on characters that are really unique in Pavetteae, i.e. the bicoloured seeds (see above) and the campanulate corolla tube (in other genera of the tribe, the tube is cylindrical or nearly so) than on the axillary inflorescences, which are not unique in the tribe - even though they are unusual, and must be discussed for this reason.

- I suggest omitting the section from "with the discovery" (line 360) to "the new Rubiaceae of Mt Gangan" (line 369) that I find rather speculative. I see no particular reason to suppose involucral cups and axillary inflorescences are plesiomorphic rather than apomorphic, both hypotheses being equally parsimonious.


Generic & specific descriptions:

- the generic diagnosis should mention three additional characters that separate Kindia from Leptactina: bicoloured seeds (cf. above), anthers not apiculate, and presence of a dense hair band near the base of the corolla tube.

- The leaves of the new species, judging from Fig. 1, are not really appressed to the substrate, as are e.g. those of Plantago media, Costus spectabilis, or Craterostigma plantagineum.

- Since the bracts are formed by the fusion of two modified leaves and their stipules, I suggest to call them "bracteal cups" for more clarity ("cupular bracts" is correct as well, but slightly less explicit).

- the description of the corolla (lines 497-505) is not adequate concerning the indumentum. The proximal portion of the tube is described as "glabrous in proximal part; middle portion of the proximal tube with a densely puberulent band 1–2 mm long". I presume this refers to the inside, but then, what about the outside? For the distal part of the tube the indumentum of both surfaces is described (but why several lines apart?) while the indumentum of the lobes is not described at all. The statement " inner surface of corolla glabrous in proximal 2.2–2.4 cm" (l. 504) is clearly in contradiction with l. 498-499 and the figures, which both make clear that there is actually a ring of hairs near the base of the tube. All this should be clarified. Please describe both the external and internal indumentum for all portions of the corolla (proximal portion of tube; distal portion of tube; lobes) and follow a consistent pattern.

- The statement that Mussaenda epiphytica seems to have lost its ability to produce long stems (lines 382-385) is, as I mentioned previously, not correct. It is indeed not usual for this species to produce long stems, but it does sometimes occur, so the ability has not been lost. I have seen it in the field on the specimen Dessein et al. 2589, which is recorded as “Epiphytic shrub or liana up to several metres long” - hence the description of the species' habit as "epiphytic shrub (sometimes becoming ± lianescent)" in Lachenaud et al., Plant Ecology and Evolution 146: 121–133 (2013). An image of the concerned specimen is available at http://mediaphoto.mnhn.fr/media/14413926262984pfqv9RVyyspcEGB . The reason why long stems are not usually produced by this species has probably more to do with limited nutrient availability in an epiphytic condition, that to them being disadvantageous.

- The authors state (lines 545-548): "It is possible that the robust, large white flowers are pollinated by a small species of bat since in June and September we saw signs of damage to the inner surface of the corolla inconsistent with visits by small insects." However, the possible involvement of larger insects (e.g. beetles, carpenter bees) is not discussed, and would seem an equally plausible hypothesis. (Whether the flowers open at night or during the day, which would be an important information in this context, is not mentioned).


Table 1

- The anther attachment is not an absolute character to separate Kindia from Leptactina, because one species of the latter genus (L. arborescens) has a sub-basal anther attachment, as in Kindia. It is an exception, however.

- As mentioned above, the difference in seed colour between the two genera should be mentioned in the table.

- The corolla tube of Leptactina cannot be described as "Only slightly campanulate" - it is not campanulate at all (see the following comment) but cylindrical, or sometimes very slightly widening at the throat.


Comments applying to various sections of the manuscript:

- The authors describe the corolla tube of Kindia gangan as "infundibular-campanulate", but "campanulate" alone would be simpler and more accurate: it means, literally, bell-shaped (this can be checked in an English dictionary), while infundibular (funnel-shaped) suggests a more gradual widening of the tube. (Since readers may not be familiar with this term, I suggest adding a brief definition in brackets the first time it is mentioned).

- The anther attachment, erroneously described as sub-apical in the abstract (l. 35) and conclusions (l. 602) is actually sub-basal, as stated in Table 1 and in the generic description.

·

Basic reporting

The authors have reworked the manuscript with great attention to the remarks of the three reviewers. They have followed most of the suggestions and remarks of the reviewers. In other cases, an acceptable rebuttal was given.

The mansucript now is correct in the comparison between Kindia and Leptactina, which was its main flaw in the previous version. The authors have kept the elaborate species description, making it in fact even longer. But it is now better understandable and the apparant contradictions in the flower size have been clarified. The elaborate description may be accepted as is.

I have some minor comments/suggestions
- Line 442 in pdf: simple (that is of unbranched hairs ...): I would delete " that is of"
- In fact, I would add to the genus description that all indumentum in Kindia is simple (with the above-mentioned sentence as explication). You could then delete all mention of simple indumentum in the species description (4 times)
- I was interested to read the author's explication on the colour pattern inside the corolla, which they claim is a sign of damage since they did not observe this pattern in all flowers. In the species description they say the colour of the corolla is white. However, the authors show a picture of a flower with dark brown spts on the inside of the corolla (which contrasts with the description). Personally, I would want the author's information on the colour pattern included in the manuscript. It would fit well within the ecology part where the authors describe bat pollination for this species: "The brown spots seem to be the result of damage by flower visitors, perhaps bats. Freshly opened flowers do not have them, nor do all flowers, only those few which show slight damage."
- In the acknowledgements the authors thank two anonymous reviewers; Please change to "three"reviewers.
-line 576 in pdf: add "by" (by a combination of characters)
- Legend of Table 1: cite also De Block & Robbrecht (1998) for the pollen characters of Leptactina

Experimental design

see first review

Validity of the findings

see first review

---

## Round 0.3 · Minor Revisions

This time, the reviewers were quite satisfied with the improvements and corrections. There are only few corrections now still to be made, and I ask you to follow them and just return the corrected version with a few comments as to whether you could fix these minor problems in a final version that will be then ready for publication. Thank you for your patience, but this contribution is important and of high standard now after such careful review and revision.

·

Basic reporting

Nothing to add from my previous review. This paper is clear, generally well-written, and forms a coherent publication. The structure of the manuscript is accurate, and so are the illustrations. The relevant literature has been cited.

Experimental design

The research presented here is certainly original, and falls within the scope of the journal. The investigation has been appropriately conducted and the methods are accurately described.

Validity of the findings

The authors' conclusion that their plant represents a new genus and species is well-supported, both by molecular and morphological data.

Additional comments

This paper is of great interest for botany, as already noted. The new version has been greatly improved from the previous ones, taking into account the reviewers' comments; when these have not been followed, the authors have provided acceptable reasons for doing so. I have only very minor modifications to suggest, which are discussed in the text of the paper. These are not important enough to justify a resubmission to reviewers.

·

Basic reporting

no comment

Experimental design

no comment

Validity of the findings

new genus fully justified

Additional comments

You still need to have a look to punctuation in the descriptions, when changing the subject you cannot use commas but need semicolons or periods

·

Basic reporting

The authors have adapted the manuscript as suggested. have no further comments except for a typo in line 530
line 530: Freshly opened flowers do not have these spot, (should be spots)

Experimental design

see previous review

Validity of the findings

see previous review

Additional comments

see previous review

---

## Round 0.4 · accepted · Accept

I am convinced that the multiple corrections and improvements have been worth while to achieve this.This is now a very important contribution of a description for a new species.

#